# Reproducibility Study: Equal Improvability: A New Fairness Notion Considering the Long-Term Impact

**Berkay Chakar**\*                                              *berkay.chakar2@student.uva.nl*

**Amina Izbassar**\*                                             *amina.izbassar@student.uva.nl*

**Mina Janićijević**\*                                           *mina.janicijevic@student.uva.nl*

**Jakub Tomaszewski**\*                                          *jakub.tomaszewski@student.uva.nl*

**Reviewed on OpenReview:** *https://openreview.net/forum?id=Yj8fUQGXXL*

## Abstract

This reproducibility study aims to evaluate the robustness of Equal Improvability (EI) - an effort-based framework for ensuring long-term fairness. To this end, we seek to analyze the three proposed EI-ensuring regularization techniques, i.e. Covariance-based, KDE-based, and Loss-based EI. Our findings largely substantiate the initial assertions, demonstrating EI's enhanced performance over Empirical Risk Minimization (ERM) techniques on various test datasets. Furthermore, while affirming the long-term effectiveness in fairness, the study also uncovers challenges in resilience to overfitting, particularly in highly complex models.

Building upon the original study, the experiments were extended to include a new dataset and multiple sensitive attributes. These additional tests further demonstrated the effectiveness of the EI approach, reinforcing its continued success. Our study highlights the importance of adaptable strategies in AI fairness, contributing to the ongoing discourse in this field of research. Our code is publicly available on GitHub.

## 1 Introduction

Throughout the last ten years, artificial intelligence (AI) has become an inseparable part of people's daily lives, penetrating every aspect of it. By being incorporated in myriads of applications not only is this technology having a rapidly increasing impact on society, but also the way people make life-changing decisions. However, it's been noted that these methods often display biases related to ethnicity, race, and gender (Casillas, 2023; Angwin, 2016). Therefore, while machine learning methods often benefit work efficiency and may help automate numerous processes, this inherent bias in AI has sparked concerns about discrimination and the need for equity.

To account for the potential biases, numerous fairness-ensuring methods have been proposed over the recent years. Some of the most prominent ones are demographic parity (DP), equal opportunity (EO), and equalized odds (EODD) (Dwork et al., 2011; Hardt et al., 2016). The general idea behind the aforementioned metrics is that they measure if the probabilities across different demographic groups are equal for each class. Nonetheless, while the majority of existing fairness metrics prove to be successful in immediate fairness, i.e. measured statically, only using the predicted class labels, they struggle with ensuring it in the long run. The notion of long-term fairness relates to a situation in which an individual puts effort into improving their features in order to become classified as a different class. A common example depicted in the literature is a person increasing their income to be qualified for a loan. While this shortcoming has already been identified in the literature e.g. Bounded Effort (BE) and Equal Recourse (ER), the existing methods have their limitations, namely vulnerability to imbalanced data or outliers for BE and ER respectively (Heidari et al., 2019;

---

\*The authors contributed equally to this work.

Gupta et al., 2019). Motivated by these substantial issues, Guldogan et al. propose Equal Improvability (EI) - an in-processing effort-based fairness technique that does not suffer from those limitations (Guldogan et al., 2023). Furthermore, the authors claim that EI ensures identical feature distributions for different groups in the long run and outperforms other methods in ensuring long-term fairness. To prove the robustness and reliability of EI they conduct various experiments and present their findings.

In this study, we aim to assess the reproducibility of the paper by replicating the experiments and further exploring various settings and scenarios to substantiate the robustness of the introduced EI.

## 2    Scope of reproducibility

The paper by Guldogan et al. (2023) investigates the issue of fairness in AI. Specifically, it addresses group fairness and proposes a new method of ensuring long-term fairness called Equal Improvability(EI). Long-term fairness is a concept in decision-making and policy that prioritizes equitable outcomes and opportunities over an extended period, rather than focusing on immediate or short-term equality. It involves considering future generations and the long-term impacts of decisions to ensure fairness and justice for all over time. In addition, group fairness is a setting where different groups are treated statistically equally. The authors introduce EI as a method for negatively classified samples to achieve an equal probability of being positively classified after improvement, regardless of which group the sample belongs to. In this study, we will verify the following claims of the original paper:

- **Claim 1:** A classifier obtained by each of the three proposed EI ensuring methods, i.e. Covariance-based, KDE-based, and Loss-based EI (See section 3.1), has a significantly smaller EI disparity value than the ERM (Empirical Risk Minimization) approach and a comparable error.

- **Claim 2:** Most existing methods have an adverse effect on long-term fairness, while EI continues to enhance it. In other words, EI accelerates the process of mitigating long-term unfairness under the dynamic scenario where the data distribution and the classifier for each demographic group evolve over multiple rounds.

- **Claim 3:** The introduced methods of achieving Equal Improvability, i.e. Covariance-based, KDE-based, and Loss-based EI, prevent an over-parametrized classifier from overfitting the data.

In the following sections of the study, we present the notion of Equal Improvability and outline the datasets, models, and hyperparameters used for evaluating its performance. In section 4 we seek to reproduce the results obtained by the authors by conducting experiments from the original paper. Moreover, we investigate the robustness of EI further by analyzing it in various scenarios and settings. Ultimately the final section of this study discusses the obtained results and evaluates the extent to which the initial claims proposed by the authors are substantiated.

## 3    Methodology

The official GitHub repository, shared by the paper authors, was found to be incomplete, encompassing code for conducting experiments only on a Synthetic dataset. Nonetheless, the initially missing code was provided by the authors upon our request. We adjusted their environmental setup to ensure compatibility with our local machines. By employing the supplied experimental settings, we have managed to replicate the majority of results illustrated in the paper. In pursuit of exploring the generalizability of the proposed method, we extend the original code and conduct additional experiments.

### 3.1    Model descriptions

**Equal Improvability (EI)** - is introduced by the authors as a novel concept in effort-based fairness that aims to balance the likelihood of acceptance for previously rejected samples across various groups, under the assumption that each rejected individual will expend a limited, specified amount of effort to improve

their chances of acceptance (Guldogan et al., 2023). It differs from other fairness metrics by focusing on the potential for improvement and equalizing acceptance rates for individuals who were initially rejected, across different demographic groups. Refer to Appendix A for the equation of EI. In addition, the authors proposed the following equation for solving the fairness-regularized optimization problem:

$$\min_{f \in \mathcal{F}} \left\{ (1 - \lambda) \frac{1}{N} \sum_{i=1}^{N} \ell(y_i, f(x_i)) + \lambda U_\delta \right\}, \tag{1}$$

They consider 3 ways of defining the penalty term $U_\delta$:

1. **Covariance-based EI penalty:** This method, inspired by Zafar et al. (2017), assesses the fairness of a score-based classifier $f$ by measuring the covariance between the sensitive attribute $z$ and the maximally improved score of rejected samples within a defined effort budget. The unfairness of a classifier $f$ is quantified by $(\text{Cov}(z; \max_k \Delta x I_k f(x + \Delta x) | f(x) < 0.5))^2$, where $I_-$ is the set of unqualified samples, and $\Delta z$ is the average sensitive attribute in this set.

2. **KDE-based Method for EI:** This approach, inspired by Cho et al. (2020), uses a Kernel Density Estimator (KDE) to approximate the probability density function of the score $f(x)$. The KDE estimates the density of the maximum achievable score by improving features within the budget $\Delta$, and the EI penalty is computed by summing the absolute difference of probability values for each group.

3. **Loss-based Method for EI:** This method calculates the fairness violation by computing the absolute difference of group-specific losses. The EI loss for a group $z$ is defined as $\tilde{L}_z = \frac{1}{|I_-; z|} \sum_{i \in I_-; z} \ell_{-1; \max_k \Delta x I_i k f(x_i + \Delta x_i)}$. This loss measures how far rejected samples in group $z$ are from being accepted after feature improvement within the budget $\Delta$.

**Models Used for Testing:** The majority of the tests were conducted using Logistic Regression (LogReg). Additionally, some tests were performed on Multilayer Perceptrons (MLPs), including:

1. A single-layer ReLU neural network with four hidden neurons.

2. A five-layer ReLU network with 200 hidden neurons per layer.

These models were employed to validate the effectiveness of the different EI methods in ensuring fairness in decision-making processes.

## 3.2 Datasets

To assess the reliability of the feature and example importance methods, the authors developed one synthetic dataset and also employed two real-world datasets, specifically the German Statlog Credit and ACSIncome-CA (Dua & Graff, 2017; Ding et al., 2021). Our study broadened this analysis by incorporating the Default of Credit Card Clients Dataset (DCC Dataset), which was selected due to its inherent gender and age bias, providing a relevant case for testing fairness in models (Yeh, 2016). Such a biased dataset allows us to evaluate the effectiveness of our methods in handling and mitigating biases. The class distribution, detailing this bias, can be found in Appendix D.

Table 1 provides a summary of the key details of these datasets. All datasets used for the experiments were split into training/test sets in the ratio of 4:1.

## 3.3 Hyperparameters

The authors of the original study provided detailed hyperparameter configurations in Appendix C.2 of their paper, as well as within the supplementary notebooks. We adhere to these specified hyperparameters for our replication efforts, ensuring consistency with the original experiments. When conducting additional experiments, we utilize similar hyperparameter settings to ensure that our results are comparable.

| Datasets | Samples | Classes | Num. of sensitive attrs. | Description |
|---|---|---|---|---|
| Synthetic | 20,000 | 2 | 1 | Synthetic data with improvable features and binary outcomes from Bernoulli/Gaussian models. |
| German Statlog Credit | 1,000 | 2 | 1 & 2 | Credit risk data from Germany, assessing creditworthiness. |
| ACSIncome-CA | 195,665 | 2 | 1 & 2 | Demographic and income data from California's ACS. |
| Default of Credit Card Clients | 30,000 | 2 | 1 | Taiwanese credit card client data, focusing on default payments. |

Table 1: Overview of the datasets used for validating the proposed methods in the label-free setting.

### 3.4 Experimental setup and code

To replicate the experiments detailed in the original paper we leverage the official codebase provided by the authors on GitHub, comprising a series of Python scripts. However, as the repository contained only the notebooks for reproducing experiments on the Synthetic Dataset, they were not compatible with both the German Statlog and ACSIncome-CA datasets due to discrepancies in hyperparameter values. Consequently, our team needed to obtain the appropriate settings by directly reaching out to the authors. Moreover, we adapted the code to support GPU computations, ultimately resulting in a substantial reduction in training time. Further alterations were made in order to evaluate the performance of EI in the scenario of multiple sensitive features. All the aforementioned modifications were implemented in Python with the codebase hosted on GitHub.

### 3.5 Computational requirements

Due to time constraints and the need to process four datasets of differing sizes, including the notably large ACSIncome-CA dataset with approximately 200,000 samples, our team conducted the experiments concurrently using both a GPU and a CPU. Specifically, we employed an NVIDIA T4 GPU and an Apple M1 Pro CPU chip for this purpose. Details regarding the runtimes and estimated energy usage are illustrated in Table 10. It should be noted that the estimated energy consumption figures are provided solely for the experiments conducted on the NVIDIA T4 GPU, consuming approximately 33Wh, i.e. 0.033kWh.

## 4 Results

In this section, we validate the main claims by replicating the experiments presented in the paper and extending them with a new experimental setup, including a new dataset and the integration of additional sensitive features. Furthermore, we check whether Equal Improvability (EI) is vulnerable to overfitting. The outcomes of our replicated experiments overall follow the same trends as the ones produced in the original paper.

### 4.1 Results reproducing original paper

**EI Disparity and Loss value check**: To verify claim 1, we compared how the 3 models, proposed by the authors, perform compared to ERM in terms of loss and EI disparity. To accomplish this we reproduced both Table 2 and Figure 3 of the original paper in which Logistic Regression (LogReg) was used as the backbone model (see the obtained Table 2 and Figure 1) (Guldogan et al., 2023). Moreover, in Table 6 we reproduced Table 3 of the original paper containing the results obtained by training a simple MLP model comprised of a single hidden layer with 4 hidden neurons (Guldogan et al., 2023). Despite identifying fractional numerical differences (see Appendix B.3 for comprehensive details), the results in the mentioned tables and figures were in general aligned with the ones illustrated in the paper. Subsequent scrutiny of these numerical deviations did not yield any solid conclusion. We suspect that the authors may have employed different seeds or hyperparameters in their experimental setup, which has not been reported either in the formal report or in

the official codebase. Alternatively, the possibility exists that certain values were reported incorrectly. We eliminated the possibility of code differences in the core algorithm as the missing notebooks were obtained directly from the authors.

| Dataset | Metric | ERM | Covariance-Based | KDE-Based | Loss-Based |
|---------|--------|-----|------------------|-----------|------------|
| Synthetic | Error Rate | .222 ± .002 | .253 ± .003 | .253 ± .009 | .246 ± .002 |
|           | EI Disp.   | .118 ± .007 | .003 ± .002 | .003 ± .003 | .002 ± .001 |
| German Stat. | Error Rate | .220 ± .009 | .262 ± .009 | .249 ± .031 | .237 ± .008 |
|              | EI Disp.   | .041 ± .008 | .022 ± .023 | **.226 ± .388** | .016 ± .013 |
| ACSIncome-CA | Error Rate | .185 ± .000 | .200 ± .000 | .196 ± .001 | .195 ± .000 |
|              | EI Disp.   | .031 ± .001 | .008 ± .001 | .005 ± .001 | .006 ± .001 |

Table 2: Error rate and EI disparities of ERM and the three proposed EI-regularized methods on Logistic Regression (LogReg). The results indicate that EI-based classifiers have a lower EI disparity without causing a significant increase in the error rate, illustrating their reduced bias towards sensitive groups.

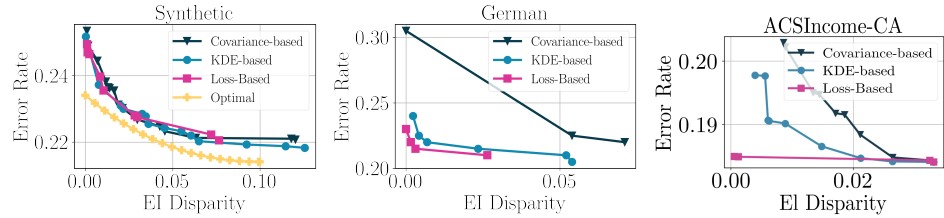

Figure 1: Tradeoff between EI disparity and error rate for Logistic Regression (LogReg). All three introduced methods, i.e. Covariance-based, KDE-based, and Loss-based EI, successfully find a tradeoff between the error rate and EI disparity, being the bottom left corner of the figure.

Yet, after a thorough examination of the obtained values we have concluded that our results substantiate claim 1, as models trained using the three introduced methods, i.e. Covariance-based, KDE-based, and Loss-based EI, perform better than the ERM on all test datasets. Since the KDE-based method is only underperforming on one of the datasets, we concluded that the authors might have used different hyperparameters for this experiment, which is why we could not reproduce their results.

**Long-term (un)fairness check:** To assess claim 2, we replicated the experiments for Figure 4 by Guldogan et al. (2023) which illustrated the persistent unfairness over time for various algorithms. Our replicated figure indicates that the EI classifier's long-term unfairness diminishes more rapidly compared to other fairness approaches, corroborating the authors' second claim. Additionally, we recreated Figure 5, observing the progression in feature distribution across three rounds of applying each algorithm. Notably, the EI method displayed a narrower gap in feature distribution between groups than other fairness concepts (DP, BE, ER, and ILFCR). Our findings for both figures align perfectly with those originally published, confirming the accuracy of our replication. This consistency strongly supports the claim regarding the EI classifier's long-term fairness effectiveness. The replicated figures can be found in Appendix B (see Figure 5).

**Robustness Check Against Overfitting**: To verify the third claim, we replicated Table 4 from Guldogan et al. (2023), which compared the Error rate and EI disparities for both the ERM baseline approach and the proposed EI ensuring methods, on an overparameterized neural network applied to the German Statlog Credit dataset. The results of our replication are illustrated in Table 3. Although the obtained EI disparities were found to be aligned with the original values, showing either exact matches or differences within a margin of one-tenth, a notable variance was observed in the error rates. Closer inspection revealed a uniform discrepancy of 0.1 across all error rates, suggesting a potential miscalculation in the originally reported values. However, it was identified that none of the methodologies, including ERM, exhibited overfitting. Consequently, the premise that EI is resistent to overfitting is rendered moot, given the baseline method (ERM) itself shows no signs of overfitting. Therefore, we deduce that the third claim lacks validity.

| Metric | ERM | Covariance-Based | KDE-Based | Loss-Based |
|---|---|---|---|---|
| Train Error | $.218 \pm .004$ | $.233 \pm .003$ | $.226 \pm .009$ | $.233 \pm .012$ |
| Test Error | $.218 \pm .010$ | $.218 \pm .010$ | $.222 \pm .007$ | $.231 \pm .007$ |
| Train EI Disp. | $.024 \pm .017$ | $.018 \pm .011$ | $.018 \pm .012$ | $.009 \pm .009$ |
| Test EI Disp. | $.064 \pm .036$ | $.050 \pm .024$ | $.070 \pm .050$ | $.062 \pm .015$ |

Table 3: Error rate and EI disparities of ERM and the three proposed EI-regularized methods on an overfit Multilayer perceptron (MLP).

### 4.2 Results beyond original paper

**Evaluating EI on a different dataset:** While the reproducibility study proves the first claim of the paper, we aim to scrutinize it further on a different dataset. That is due to the data in both datasets, utilized in the original paper, i.e. German Statlog Credit and ACSIncome-CA, having considerable biases towards some of its sensitive groups (see Appendix D). To this end, we employ the Default of Credit Card Clients Dataset, containing a substantially lower bias for the *Sex* feature, as illustrated in Appendix D. We trained the Logistic Regression model for 100 epochs, with a setting of a single sensitive and improvable feature being *Sex* and *Education* respectively. Moreover, after tuning the hyperparameters we set the batch size to *512* and the learning rate to *0.01*.

By analyzing the obtained results, depicted in Table 4 and Figure 6, we indeed note a smaller EI Disparity for all proposed EI ensuring methods, i.e. Covariance-based, KDE-based and Loss-based EI. Therefore, as we did not observe any unusual divergences, we conclude that the experiment supports claim 1.

| Metric | ERM | Covariance-Based | KDE-Based | Loss-Based |
|---|---|---|---|---|
| Error Rate | $.175 \pm .001$ | $.177 \pm .001$ | $.176 \pm .001$ | $.176 \pm .001$ |
| EI Disp. | $.009 \pm .001$ | $.004 \pm .002$ | $.002 \pm .001$ | $.003 \pm .001$ |
| DP | $.020 \pm .001$ | $.017 \pm .001$ | $.016 \pm .001$ | $.014 \pm .001$ |
| EO | $.027 \pm .001$ | $.018 \pm .001$ | $.016 \pm .001$ | $.009 \pm .003$ |
| EODD | $.027 \pm .001$ | $.018 \pm .001$ | $.016 \pm .001$ | $.009 \pm .002$ |

Table 4: Results of evaluating ERM and the proposed EI ensuring methods on the Default of Credit Card Dataset using Logistic Regression (LogReg). The results indicate that EI-based classifiers have a lower EI disparity without causing a significant increase in the error rate, illustrating their reduced bias towards sensitive groups.

**Extending the experiments in the paper by adding another sensitive feature into consideration** – to test whether the three methods proposed by the author still outperform ERM in achieving EI Fairness in the case when multiple sensitive features have to be accounted for, we have decided to run the experiments on the error rate/EI Disparity tradeoff with an additional sensitive feature for German Statlog Credit, ACSIncome-CA and Default of Credit Card Clients datasets. For that, we used the formula for each method of defining the loss function, and instead of having a single penalty term U, we summed the penalty terms for all sensitive features and multiplied them by the weighting factor. Hence the loss becomes:

$$\min_{f \in F} \left( (1 - \lambda) \frac{1}{N} \sum_{i=1}^{N} \ell(y_i, f(x_i)) + \lambda \sum_{z=z_1}^{z_n} U_{\delta z} \right) \tag{2}$$

For the German Statlog Dataset in addition to the "age" feature, the "sex" feature was used for the multiple sensitive feature experiment. Similarly, we extend the sensitive feature set of the ACSIncome-CA Dataset to contain not only "sex" but also "age". We assume all sensitive features to be binary, hence we split the "age" feature using a cutoff of 30 and encoding all values less than 30 as 0, and the other ones as 1. Since both features were already labeled as sensitive in the other dataset, we concluded that they were appropriate

choices for the multiple sensitive feature experiment. The data biases for both *Sex* and *Age* groups are presented in Appendix D, for each dataset separately.

Results of the tradeoff experiments for the German Statlog Dataset represented in Table 9 suggest that the Covariance-based method achieves a similar accuracy as ERM with a slightly lower EI disparity, whereas KDE based method has a drop of accuracy by 0.059 and the decrease in EI Disparity by 0.057. The Loss-based method for the setting with multiple sensitive features resulted in increased disparity, which contradicts the claim. We suspected that it was due to the group imbalance after splitting the dataset, and we tried to mitigate that by shuffling the dataset before splitting it. This did not yield any substantial improvement, hence, we concluded that this is the result of a conflict between the sensitive features. The algorithms may discontinue converging at a point where the disparity for one sensitive feature cannot be further improved without increasing the disparity between groups of another sensitive feature. From the results of the experiment, we infer that claim 1 only holds for KDE and Covariance-based methods.

**Testing the performance of EI in promoting long-term fairness for the case of multiple sensitive features** – to validate claim 2 for the case of multiple sensitive features, we ran the experiment when there were two sensitive features (each having two groups). To account for the disparity values of multiple sensitive features, instead of looking for thresholds that minimize the disparity concerning one sensitive feature, we add the disparities between groups for all sensitive features and seek thresholds that minimize this sum. By minimizing the sum we account for the overall disparity across sensitive features. Even though doing so might increase the disparity for one of the sensitive features, it would mean that the disparity for the other sensitive feature (that has been much larger), was decreased, effectively decreasing the sum of disparities and resulting in lower overall disparity. We considered four scenarios of group distributions given sensitive features, where each sensitive feature had 2 groups: (i) $z1$: (0, 1), (1, 0.5), $z2$: (0, 0.5), (1, 1); (ii) $z1$: (0, 0.5), (1, 1), $z2$: (0, .5), (1, .5); (iii) $z1$:(0, .5), (1, .5), $z2$: (0, 2), (0, 1); (iv) $z1$: (0, 2), (0, 1), $z2$: (0, 1), (1, 0.5). Figure 7 illustrates the evolution of the two groups concerning a sensitive feature, when different algorithms are applied. Though the results for individual distributions are slightly inferior to those in the original paper, considering that there are multiple sensitive features concerning which the group distance has to be minimized, it was expected. Moreover, EI still results in the lowest distance for almost all distributions in t=3 iterations. Figure 2 shows the change of distance between groups over several iterations. For both sensitive features, EI results in the lowest disparity for all distributions, supporting claim 2.

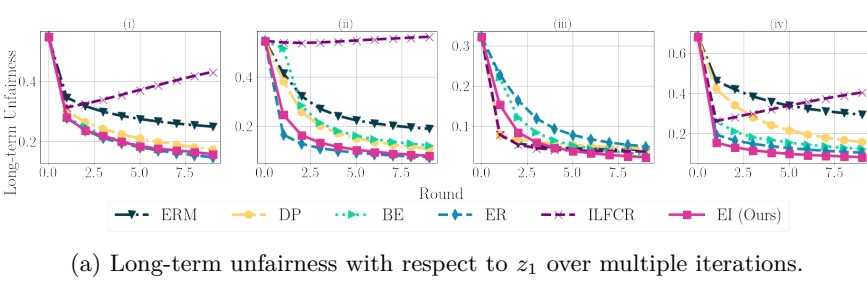

(a) Long-term unfairness with respect to $z_1$ over multiple iterations.

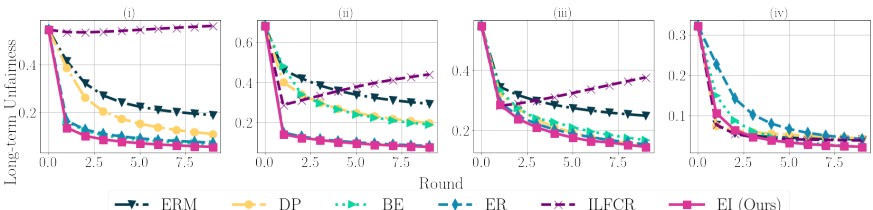

(b) Long-term unfairness with respect to $z_2$ over multiple iterations.

Figure 2: Long-term unfairness over multiple iterations for different sensitive features. The figures illustrate that the disparity between the sensitive group probability distributions reduces faster for the EI classifier than for the other metrics, indicating that it is more favourable for achieving long-term fairness.

**Overfitting robustness check with a more complex model**: in order to verify claim 3, we trained a complex Multi-Layer Perceptron (MLP) on a subset of the ASC-Income dataset, specifically comprising the first 1000 samples. Given the limited size of the subset, we opted for an MLP with 10 hidden layers, each consisting of 800 hidden neurons, expecting it to exhibit overfitting. Analyzing the results, depicted in Table 5, indicated that both the KDE-Based and Loss-Based method exhibit clear overfitting, with training accuracies and EI disparities being significantly lower than the test ones. Therefore, the proposed methods cannot be substantiated as resistant to overfitting on an overparametrized model.

| Metric | ERM | KDE-Based | Loss-Based |
|---|---|---|---|
| Train Error | .079 ± .012 | .094 ± .014 | .126 ± .015 |
| Test Error | .221 ± .008 | .238 ± .023 | .223 ± .015 |
| Train EI Disp. | .010 ± .005 | .016 ± .014 | .008 ± .006 |
| Test EI Disp. | .054 ± .009 | .050 ± .034 | .024 ± .014 |

Table 5: Error rate and EI disparities of ERM and the proposed EI-regularized methods on an overparameterized Multilayer perceptron (MLP) using a subset of ACS-Income dataset. The error rate and EI disparity values for all methods are indicative of overfitting, with considerably lower error rate and EI disparity for the train set compared to the test set.

## 5 Discussion

In this study, we conducted multiple experiments to replicate and validate the key findings from the original research on Equal Improvability (EI). Our efforts generally confirm the original claims, successfully reproducing most of the initial findings across various datasets and models. We examined three primary claims made by the authors, finding them largely substantiated, with some exceptions in specific scenarios.

During our investigation, we encountered minor discrepancies, especially when introducing additional sensitive features and using different datasets. To address these inconsistencies and gain further insights, we reached out to the original authors for clarification and additional resources. They promptly provided support, which helped us understand the nuances better, though some results remained challenging to replicate precisely.

Additionally, we explored the question of the EI approach's robustness against overfitting. The results were not consistent; while EI performed well under certain conditions, it did not consistently demonstrate immunity to overfitting, particularly with complex models. This highlighted the need for cautious application and potential enhancement of the EI method in environments susceptible to overfitting.

To further test the robustness and applicability of the EI approach, we expanded our experiments with the Default of Credit Card Clients Dataset and found that the EI approach still held up well, reinforcing the validity of the original conclusions. Moreover, we conducted experiments with the integration of multiple sensitive features, providing a more comprehensive assessment of the EI method's efficacy in diverse scenarios.

To conclude, our study not only confirms the effectiveness of the EI approach but also sheds light on its limitations. While EI holds promise for enhancing AI fairness, our findings highlight the need for further research into more adaptable approaches that can effectively address the diverse challenges of AI applications.

### 5.1 What was easy

*Communication with authors:*

The communication with the authors of the study was notably easy and efficient, which greatly facilitated the reproduction process. The authors consistently provided prompt responses, ensuring that any queries or clarifications we needed were addressed quickly. Refer to section 5.3.

*Reproducing results for synthetic dataset:*

The reproduction of results for the synthetic dataset was remarkably smooth and efficient, largely due to the immediate availability of all necessary files. This complete access enabled a straightforward replication process. Additionally, the results we achieved closely aligned with those published in the paper, reflecting the study's reproducibility and reliability for this dataset. Furthermore, the experiments benefited from a notably short runtime, which significantly streamlined our overall effort in this part of the study.

*Clarity and Quality of the Paper:*

The paper was well-written, concise, and successful in clearly conveying its general message. This ease of understanding greatly assisted in grasping the study's concepts and objectives, contributing to a smoother reproduction process.

## 5.2   What was difficult

*Environment Setup and Dependency Issues:* The initial challenge was in setting up the required environment. The provided requirements file had incorrect dependencies, which hindered the creation of a compatible environment. We had to revise and adjust the dependency versions to ensure their mutual compatibility, which was more time-consuming than anticipated.

*Dataset-Specific Hyperparameters:* A significant issue was the lack of clarity regarding hyperparameters used for different datasets, namely the Income and German Datasets. The original paper and the accompanying GitHub repository did not specify distinct hyperparameters for each dataset, leading to initial difficulties in reproducing the results. Through direct communication with the authors, we obtained the missing code and hyperparameters. Although we were unable to replicate the exact numerical results, the overall conclusions of the study were consistent with the authors' claims.

*Adaptation for Table 4 Reproduction:* In our attempts to reproduce Table 4, we faced a hurdle when adjusting the number of weights for a layer. The method for calculating optimal effort became inapplicable, and guidance for adjustments was absent in both the code and the paper. This omission required us to develop an understanding and approach independently, illustrating the need for more detailed documentation in research.

*Code Limitation for Specific Scenarios:* Another challenge was the limitation of the provided code, which was tailored for a specific scenario involving data with a single binary sensitive feature. To address this, we expanded the scope by adapting the code for multivariate cases, allowing it to handle multiple binary sensitive features. This extension enabled us to conduct further experiments to assess whether EI (Equality of Impact) outperforms other fairness methods in more complex settings. This adaptation highlighted the necessity for more versatile and inclusive code in research studies.

## 5.3   Communication with original authors

Initially, we contacted the authors to inquire about the hyperparameters used. They promptly responded within a day, guiding us on the appropriate parameters and providing the Logistic Regression (LR) notebooks for the Income and German datasets. However, notebooks detailing the tradeoff analysis were absent. Following their instructions, we created these notebooks ourselves, but our results diverged from those reported in the original study. This led to a second communication with the authors, who quickly supplied the missing tradeoff notebooks for both datasets. Using these notebooks, our subsequent experiments produced patterns similar to those in the original paper, yet with some variances. The authors have acknowledged these discrepancies and expressed their intention to update the repository, which we believe may resolve these issues. As of this report, we are awaiting this update. Furthermore, we noted the absence of Multi-Layer Perceptron (MLP) files needed to reproduce two tables from the article. Upon requesting these from the authors, they promptly sent us the required files. In total, we contacted the authors thrice, each time marked by their swift and helpful responses. Despite their assistance, replicating the exact results for the Income and German datasets remains a challenge. We are optimistic that the upcoming repository update will aid in resolving these discrepancies.

## 6   Aknowledgements

We would like to thank Alexandre da Silva Pires for his supervision and valuable support during the development of this work.

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

# A   Appendix

## A   Equal Improvability equation

In the work of Guldogan et al. (2023), the principle of Equal Improvability is formulated through the following equation:

$$P\left(\max_{\mu(\Delta x)\leq\delta} f(x+\Delta x) \geq 0.5 \mid f(x) < 0.5, z = \bar{z}\right) = \mathbb{P}\left(\max_{\mu(\Delta x)\leq\delta} f(x+\Delta x) \geq 0.5 \mid f(x) < 0.5\right)$$

f(x) < 0.5 indicates samples labeled as 0 by the classifier, and f(x) ≥ 0.5 indicates that the sample was labeled as 1. The formula for EI ensures that samples belonging to different groups z that were classified as 0 are equally likely to be classified as 1 after applying effort. The formula for $\mu$ can vary for different cases.

## B   Reproducibility results

### B.1   Error rate and EI disparities of ERM and the three proposed EI-regularized methods on MLP

| Dataset | Metric | ERM | Covariance-Based | KDE-Based | Loss-Based |
|---------|--------|-----|------------------|-----------|------------|
| Synthetic | Error Rate | .205 ± .003 | .244 ± .007 | .227 ± .009 | .226 ± .009 |
|  | EI Disp. | .140 ± .036 | .004 ± .002 | .010 ± .005 | .012 ± .009 |
| German Stat. | Error Rate | .221 ± .010 | .299 ± .012 | .241 ± .022 | .238 ± .035 |
|  | EI Disp. | .059 ± .046 | .013 ± .025 | .045 ± .058 | .013 ± .019 |
| ACSIncome-CA | Error Rate | .181 ± .002 | .202 ± .002 | .184 ± .002 | .185 ± .002 |
|  | EI Disp. | .034 ± .001 | .012 ± .010 | .008 ± .004 | .003 ± .002 |

Table 6: Error rate and EI disparities of ERM and the three proposed EI-regularized methods on Multilayer perceptron (MLP). The results indicate that EI-based classifiers have a lower EI disparity without causing a significant increase in the error rate, illustrating their reduced bias towards sensitive groups.

### B.2   Long-term (un)fairness check:

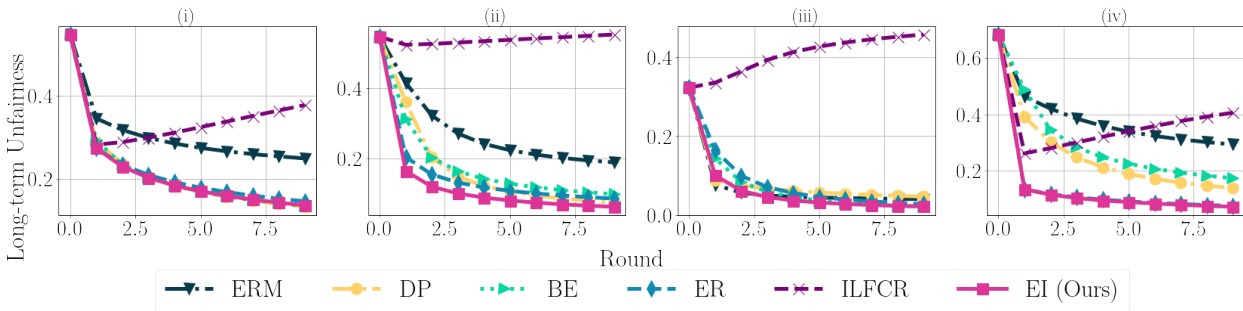

Figure 3: Long-term unfairness at each round $t$ for various algorithms.

Figure 4: Long-term unfairness over multiple iterations. The figure illustrates that the disparity between the sensitive group probability distributions reduces faster for the EI classifier than for the other metrics, indicating that it is more favourable for achieving long-term fairness.

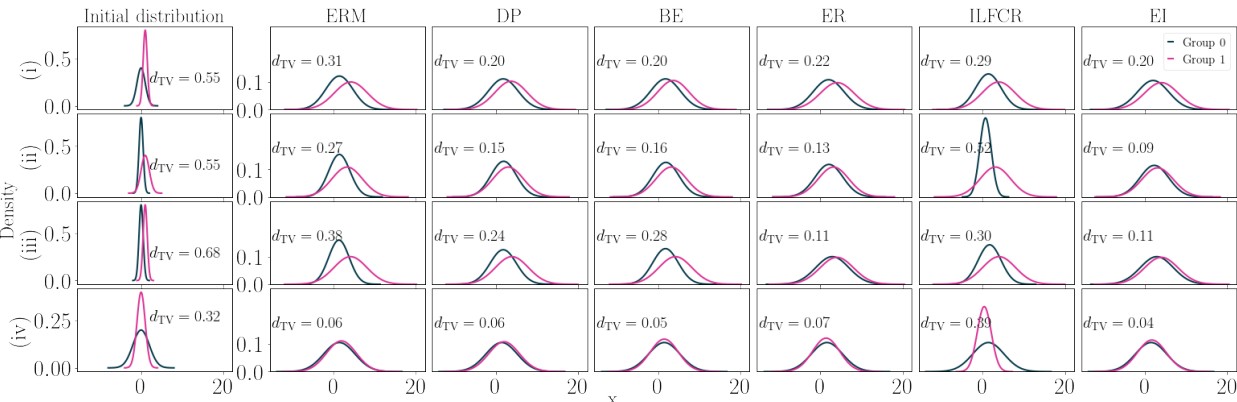

Figure 5: Probability distribution for each sensitive group after updating improvable features using each algorithm for 3 iterations. In comparison to current fairness notions, such as Demographic Parity (DP), Bounded Effort (BE), Equal Recourse (ER), and Individual-Level Fair Causal Recourse (ILFCR), EI results in a reduced gap in feature distribution across different groups.

### B.3 Numerical discrepancies in Error rate and EI disparities

In the tables below we depict the numerical discrepancies in Error rate and EI disparities of ERM and the three proposed EI-regularized methods for a Logistic Regression and Multilayer Perceptron. We present it as a difference between the values obtained by the team and the values reported in the original paper.

We suspect that the numerical variations could stem from the authors using varying seeds or hyperparameters in their experimental framework, a detail not mentioned in either the formal report or the official code repository.

| Dataset | Metric | ERM | Covariance-Based | KDE-Based | Loss-Based |
|---|---|---|---|---|---|
| Synthetic | Error Rate | .001 ± .001 | .000 ± .000 | .003 ± .006 | .000 ± .001 |
| | EI Disp. | .001 ± .000 | .001 ± .000 | -.002 ± .000 | .000 ± .000 |
| German Stat. | Error Rate | .000 ± .000 | .000 ± .000 | .006 ± .007 | .000 ± .000 |
| | EI Disp. | .000 ± .000 | .001 ± .004 | **.191 ± .362** | .001 ± .004 |
| ACSIncome-CA | Error Rate | .001 ± .000 | .000 ± .000 | .000 ± .001 | .002 ± .000 |
| | EI Disp. | .000 ± .000 | .000 ± .000 | .000 ± .000 | .000 ± .000 |

Table 7: Logistic Regression (LogReg)

| Dataset | Metric | ERM | Covariance-Based | KDE-Based | Loss-Based |
|---|---|---|---|---|---|
| Synthetic | Error Rate | .000 ± .000 | .002 ± .001 | .000 ± .001 | -.003 ± -.003 |
| | EI Disp. | -.001 ± .000 | .000 ± .000 | -.001 ± -.001 | -.006 ± .000 |
| German Stat. | Error Rate | .000 ± .000 | .000 ± .000 | .009 ± .004 | .000 ± .000 |
| | EI Disp. | .000 ± .001 | .000 ± .000 | .004 ± .033 | .000 ± .000 |
| ACSIncome-CA | Error Rate | .000 ± .000 | .000 ± .000 | .002 ± .000 | .000 ± .001 |
| | EI Disp. | -.008 ± -.001 | .002 ± .004 | -.002 ± .002 | -.003 ± -.001 |

Table 8: Multilayer perceptron (MLP)

## C Extended analysis

| Dataset | Metric | ERM | Covariance-Based | KDE-Based | Loss-Based |
|---|---|---|---|---|---|
| German Statlog | Error Rate | 0.214 ± 0.015 | 0.214 ± 0.016 | 0.273 ± 0.034 | 0.295 ± 0.018 |
| | EI Disparity | 0.121 ± 0.037 | 0.120 ± 0.040 | 0.064 ± 0.087 | **0.400 ± 0.490** |
| ACSIncome | Error Rate | 0.184 ± 0.000 | 0.200 ± 0.000 | 0.196 ± 0.000 | 0.194 ± 0.000 |
| | EI Disparity | 0.031 ± 0.001 | 0.008 ± 0.001 | 0.005 ± 0.001 | 0.006 ± 0.002 |
| Default of Credit Card Clients | Error Rate | 0.174 ± 0.001 | 0.177 ± 0.001 | 0.175 ± 0.001 | 0.191 ± 0.003 |
| | EI Disparity | 0.013 ± 0.002 | 0.003 ± 0.001 | 0.003 ± 0.000 | 0.003 ± 0.002 |

Table 9: Error rate and EI disparities of ERM and the three proposed EI-regularized methods on Logistic Regression (LogReg) trained using two sensitive features. he results indicate that EI-based classifiers have a lower EI disparity without causing a significant increase in the error rate, illustrating their reduced bias towards sensitive groups. However, the Loss-based classifier failed to achieve a lower EI disparity value than ERM classifier.

## C.1 Results of evaluating EI on the Default of Credit Card Clients Dataset

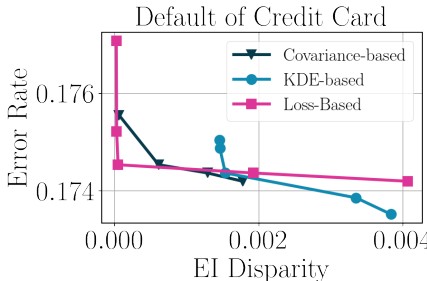

Figure 6: Tradeoff between EI disparity and error rate for the Default of Credit Card Dataset using Logistic Regression (LogReg). All three introduced methods, i.e. Covariance-based, KDE-based, and Loss-based EI, successfully find a tradeoff between the error rate and EI disparity, being the bottom left corner of the figure.

## C.2 Long-term (un)fairness evolution with multiple sensitive features

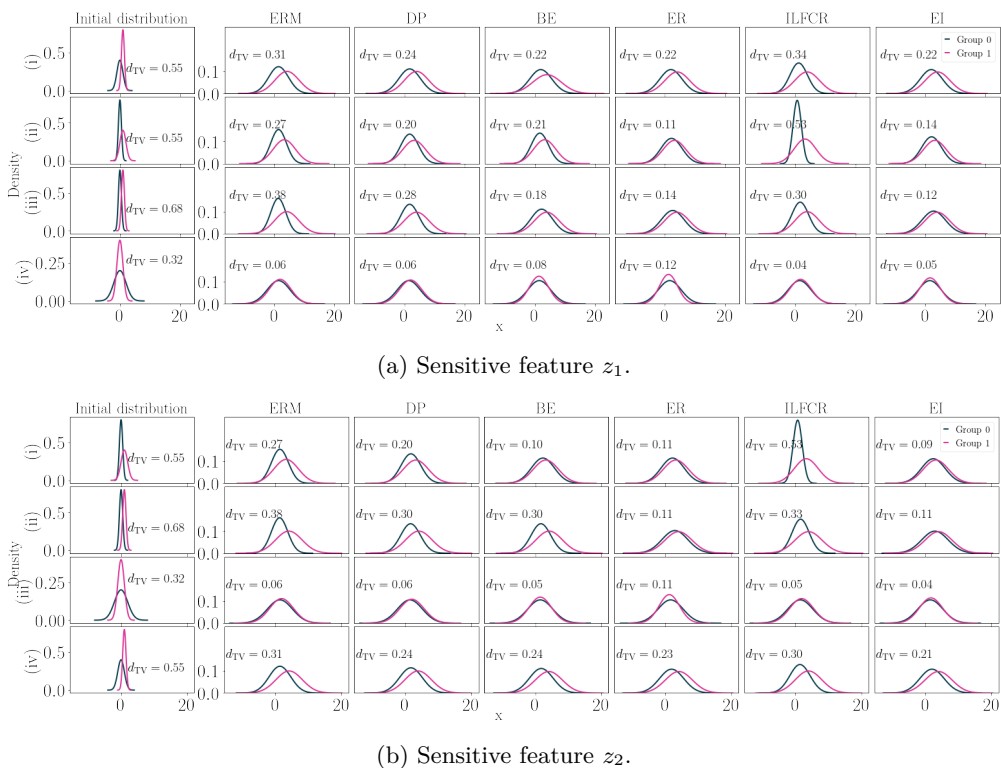

Figure 7: Probability distribution for each sensitive group after updating improvable features using each algorithm for 3 iterations. In comparison to current fairness notions, such as Demographic Parity (DP), Bounded Effort (BE), Equal Recourse (ER), and Individual-Level Fair Causal Recourse (ILFCR), EI results in a reduced gap in feature distribution across different groups.

## D    Bias in the utilized datasets

### D.1    Gender bias

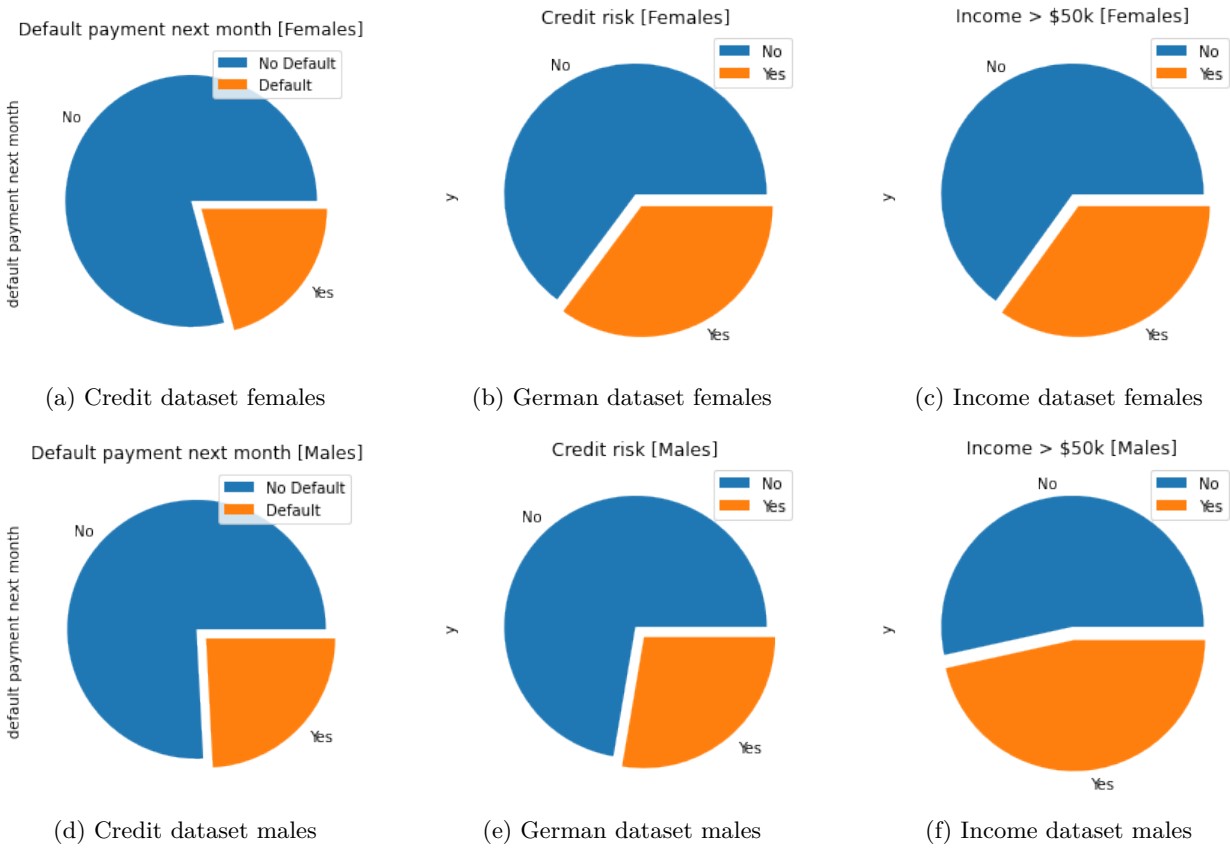

Figure 8: Gender bias in various datasets

## D.2 Age bias

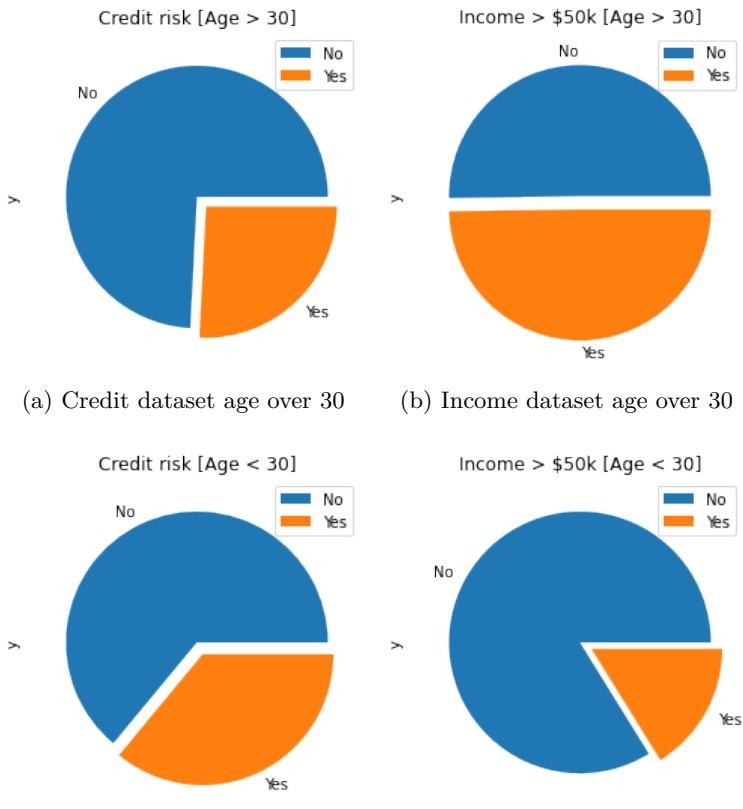

(a) Credit dataset age over 30  (b) Income dataset age over 30

(c) German dataset age below 30  (d) Income dataset age below 30

Figure 9: Age bias in various datasets

## E   Training runtimes and energy consumption for the conducted experiments

We strive to calculate the impact of our experiments on the environment. To this end, we calculate the carbon footprint using the following formula.

$$CO_2e = CI * PUE * Power * time \tag{3}$$

For running the experiments we employed an NVIDIA T4 GPU, being available at Google's data center facilities. As those include servers in various locations in the United States, e.g. Oregon and Iowa in our case, to calculate the total carbon emission we take advantage of the average PUE (Power Usage Effectiveness) reported by Google being 1.10 and the approximated carbon intensity (CI) of 0.86 pounds of CO2 emissions per kWh (Google, 2023; Administration, 2023). Based on these figures, the total carbon emission was calculated to be approximately 30.708 *lb. $CO_2e$* i.e. 13.929 *kg $CO_2e$*. See Table 10 for all runtime details.

| Dataset | Experiment | Model | Sensitive attrs. | Device | Runtime | Est. EC |
|---|---|---|---|---|---|---|
| Synthetic | Training | LogReg | 1 | CPU | 12 min | - |
| Synthetic | Tradeoff | LogReg | 1 | CPU | 36 min | - |
| German Stat. | Training | LogReg | 1 | CPU | 2 min | - |
| German Stat. | Tradeoff | LogReg | 1 | CPU | 5 min | - |
| German Stat. | Training | LogReg | 2 | CPU | 3 min | - |
| German Stat. | Tradeoff | LogReg | 2 | CPU | 7 min | - |
| German Stat. | Training | MLP | 1 | CPU | 2 min | - |
| German Stat. | Training | DNN | 1 | CPU | 5 min | - |
| ACSIncome-CA | Training | LogReg | 1 | GPU | 3.5 h | 0.116 kWh |
| ACSIncome-CA | Tradeoff | LogReg | 1 | GPU | 6 h | 0.198 kWh |
| ACSIncome-CA | Training | LogReg | 2 | GPU | 5 h | 0.165 kWh |
| ACSIncome-CA | Tradeoff | LogReg | 2 | GPU | 6.5 h | 0.281 kWh |
| ACSIncome-CA | Training | MLP | 1 | GPU | 6.5 h | 0.215 kWh |
| Default of Credit Card Clients | Training | LogReg | 1 | GPU | 0.22 h | 0.007 kWh |
| Default of Credit Card Clients | Tradeoff | LogReg | 1 | GPU | 2.3 h | 0.076 kWh |
| Default of Credit Card Clients | Training | LogReg | 2 | GPU | 0.34 h | 0.0112 kWh |

Table 10: Training runtimes on various datasets accompanied by GPU energy consumption estimates.

