# OpenReview forum: "Reproducibility Study: Equal Improvability: A New Fairness Notion Considering the Long-Term Impact"
_TMLR — Accepted by TMLR_

### Review · Reviewer_Bho8 · 2024-04-12

**Summary Of Contributions:**

This work evaluates the robustness of Equal Improvability (EI), which is an effort-based framework for ensuring long-term fairness. To do so, the authors analyze three EI-regularization techniques, covariance-based, KDE-based, and loss-based. The findings largely agree with previous results that these techniques provide enhanced performance over normal empirical risk minimization (ERM). Furthermore, the authors uncover that EI ensuring methods are not always robust to overfitting. Lastly, the authors evaluate EI on a new dataset and extend previous experiments by adding other sensitive attributes. In conclusion, the study confirms that EI is effective find that the primary claims made by the authors of the original work are substantiated by the results produced in this work.

**Audience:**

Yes

**Broader Impact Concerns:**

No concerns.

**Claims And Evidence:**

Yes

**Requested Changes:**

- In the 2nd paragraph on page 7, z1 and z2 should be written in math mode.

**Strengths And Weaknesses:**

Strengths
- The authors comprehensively evaluated the methods/results of the paper of Guldogan et al. 2023.
- The authors were able to validate many of the claims in the original paper and if they could not they highlighted what the discrepancies were exactly.
- The authors did additional experiments to evaluate the pros/cons of the EI.

Weaknesses:
- I do not see any major weaknesses in this paper.

---

> ### Author Response · Authors · 2024-05-31
>
> Thank you for your review. We addressed the point you raised. We appreciate your contribution.

---

### Review · Reviewer_QzBZ · 2024-04-18

**Summary Of Contributions:**

In general, this paper conducts reproducibility study of the paper "Equal Improvability: A New Fairness Notion Considering the Long-Term Impact", especially three claims: (i) the three proposed EI learning methods can reduce EI disparity (especially when comparing to ERM), (ii)  EI accelerates the process of mitigating long-term fairness under the dynamic scenario, and (iii) their proposed approaches do not cause overfitting when the model is over-parameterized.

To verify the three claims, the authors communicated with the authors of that paper, and run the corresponding experiments to verify whether the results provided by the paper is reproducible or not. In addition to that, the author also tried experiments beyond the original paper, including evaluating the proposed methods on a different dataset, and extending the experiments in the paper by considering the scenario of multiple sensitive features.

**Audience:**

Yes

**Claims And Evidence:**

No

**Requested Changes:**

- [Major] Please provide more details of the experiment setting.
    - For the results that lead to a strong conclusion (especially when fail to reproduce the consistent results as the original paper), it is important to report the detailed experiment setting to help reviewer understand what is going on.
    - Does the equation 2 mean that you simply adding the penalty obtained from each sensitive attribute? I want to suggest to combine multiple sensitive attributes into one, and this is what most of the existing fair learning papers do. To be more specific, for instance, assume two sensitive attribute: gender (female & male), and race (white & black), we can combine them into one attribute named sensitive (white female & black female & white male & black male), and then we should still only have one penalty term.
- [Medium] The choice of claim 3 is a bit random and lack of justification. While the results seem to be helpful, more details about the experiment setting is needed to verify the reliability of the conclusion. Moreover, it would be great to add experiments on EI v.s. BE and EI v.s. ER.
- [Minor] Fixing the typos and cleaning up the draft is needed.

**Strengths And Weaknesses:**

Strength:

- [Major] Authors understand well about the original paper, and made effort to communicate with the authors of original paper. Overall, this paper is a relatively complete reproducible study.
- [Major] In general, this paper is well-written. But as I mentioned below, there are still many places can be improved.

Weakness:

- [Major] Many details of the experiment setting are missing, thus hard to justify whether the settings are reasonable or not and whether the conclusions are correct or not. For instance, How did you extend the experiments into multiple sensitive feature case? Is there any code? The equation 2 is not very informative, and seem not the correct way of extending their experiments into multiple sensitive feature case (we can discuss more if the authors can provide more detailed explanation). Moreover, why the results of the existing experiments are different from the original paper. Do you use different seed?
- [Medium] Many decisions made by this paper is not well justified. For instance, the third claim listed in the paper does not seem to be a major claim of the EI paper. Why do authors choose this claim? In contrast, there are other interesting claims which plays a more important role in their paper. For instance, to motivate their proposed fairness notion, they pointed out the problems of similar fairness notions such as ER and BE. Replicating their experiments or trying some other possible reasonable data distributions and see whether the conclusions they made still hold will be more interesting.
- [Minor] There are some typos in the manuscript. Many links are broken, too. Here are some issues that I found, and there may exist more.
    - Five lines above the Sec. 3, the link of “section 4” is broken.
    - The citation style sometimes are incorrect. For instance, in the fourth line under Sec 3.1, it should be “chance of acceptance (Guldogan et al., 2023).” instead of “chance of acceptance.  Guldogan et al. (2023).” There are many such issues, which could be fixed by using commands like `\citep` instead of `\cite` or `\citet`.
    - What is the meaning of“[NEW]” for the paragraph “Robustness Check Against Overfitting” in page 5?
    - The bottom line of Table 4 is missing.
    - In Table 4, the “DP,” “EO,” “EODD” should be “DP Disp.,” “EO Disp.,” and “EODD Disp..”
    - Link of “section 6.4” in the last second line of page 8 is broken.
- [Minor] The clarify of this paper can be improved. For instance, in the table 1, there is a column "sensitive attrs.", but it is not very clear what does "1", "2", and "1&2" mean. Meanwhile, many notations in the paper are not explained at all (but it is not a huge problem since authors are using exactly the same notations as the original paper, which has been explained well).

---

> ### Author Response · Authors · 2024-05-31
>
> Thank you for the feedback. We have addressed your remarks below:
>
> 1. Detailing Experiment Settings:
> We appreciate your observation regarding the detail of the experimental settings. We have outlined the specifics of our experiments in the subsequent sections:
> - 3.2 Datasets
> - 3.3 Hyperparameters: The original study's authors have provided a comprehensive description of the hyperparameter configurations in Appendix C.2 of their paper, along with supplementary notebooks.
> - 3.4 Experimental Setup and Code: We have built upon the official code base provided by the authors. Our repository is available at https://anonymous.4open.science/r/ei_fairness_reproducibility-BEF1. The code base includes all the notebooks used in our experiments, ensuring they can be easily reproduced and verified.
>
> 2. Concerns Regarding Equation (2):
> Thank you for pointing out your concerns with Equation 2. We acknowledge that there are several methods to aggregate unfairness across multiple sensitive attributes. However, we find the approach discussed in the paper to be the most appropriate. The method you suggested, which combines different sensitive features into one, increases the dimensionality. For instance, combining just two attributes—gender and race—yields four combinations. Should more attributes be included, the combinations would increase exponentially, e.g., 10 sensitive features, each with 5 unique values, would result in 5^10 groupings, compared to the manageable number in our proposed method being 5*10.
>
> 3. Variations in Experimental Results:
> In Section 4.1, "Results Reproducing the Original Paper," we explore the potential reasons for discrepancies in the experimental outcomes. We consider the possibility that the original authors may have used different seeds or hyperparameters, which were not reported. Additionally, we can't rule out the reporting errors in the original documentation. We have accessed the missing notebooks directly from the authors, eliminating any discrepancies in the core algorithm. Despite experimenting with various seeds, including those provided by the original authors, we have yet to achieve results that align perfectly with those in the original paper.
>
> 4. Selection of Specific Claims for Verification:
> The choice to verify the claim related to overfitting was driven by its relevance to ongoing challenges in machine learning. The original paper posits that the methods introduced for achieving Equal Improvability—namely, Covariance-based, KDE-based, and Loss-based—are robust against overfitting. This claim intrigued us due to its potential impact on the field, prompting us to examine its validity in our study. We have included the notebooks with experiments in the repository: https://anonymous.4open.science/r/ei_fairness_reproducibility-BEF1
>
> 5. Addressing Typos and Reference Issues:
> We are grateful for your attention to detail regarding the typos and broken references in our manuscript. We have corrected these in the revised version of our paper and thank you for your constructive feedback.

---

> > ### Comment · Reviewer_QzBZ · 2024-06-17
> > **Thanks for your responses.**
> >
> > My concerns are addressed. Thanks the authors for answering my questions.

---

### Review · Reviewer_nWRX · 2024-05-17

**Summary Of Contributions:**

The paper performs a reproducibility study of the paper "Equal Improvability: A New Fairness Notion Considering the Long-term Impact" by Guldogan et al. (2023). The original paper proposes a new group fairness metric called "Equal Improvability" (EI) which aims to equalize the probability of receiving a positive outcome for a sample after improving its features over time. This EI fairness notion aims to take into account the dynamic nature of the fairness in decision making systems compared to static group fairness metrics. The original paper shows improvement of models trained to enforces EI vs ERM with minimal loss of accuracy. The paper also shows slight improvement over existing similar dynamic fairness metrics, namely "Equal Recourse" (ER, Gupta et al. 2019) and Bounded Effort (BE, Heidari et al. 2019).

In this reproducibility study, the authors assess the claims of the original paper pertaining to improvement of ERM in achieving EI as well as the long term properties of an EI-trained model. They also assess the overfititng properties of these EI-based training strategies. The main contributions are in (1) reproducing the results from Guldogan et al. (2023) on the same datasets (1 synthetic, 2 real) as in the original paper and on one additional real dataset, (2) extending the method in Guldogan et al. (2023) from 1 sensitive attribute to 2 sensitive attributes, and (3) assessing the robustness to overfitting using a more complex model than in the original paper (MLP vs logistic regression).

**Audience:**

Yes

**Claims And Evidence:**

Yes

**Requested Changes:**

As specified above, I would like to see additional experiments comparing EI and BE and ER and a discussion of settings where EI provides an improvement over these comparative methods.

**Strengths And Weaknesses:**

Strengths:

The paper is well written. The claims from the original paper being assessed are well described and the contributions are well stated. As a reproducibility study, the authors made an effort go beyond the experiments in the original paper and explore the performance of the model on a new dataset as well as in a different setting (from a single sensitive attribute to multiple attributes). The results are well explained in cases where their is a discrepancy with the original paper.

Weaknesses:

A limitation in the experiment resides in mainly comparing the 3 training strategies for enforcing EI with ERM. While this is a setting in the paper, interesting results in the original paper are in the comparison with similar dynamic fairness metric (BE and ER). Indeed, the Figures 4 and 5 in the original paper which compare to BE and ER show minimal improvement of EI compared to those methods. In the reproducibility study, I would have appreciated more comparison with those methods and an assessment of when EI provides substantial improvement. This can also be seen in the experiments with multiple attributes.

---

> ### Author Response · Authors · 2024-05-31
>
> Thank you for your valuable feedback. We do believe that the study could benefit from such a comparison, although the main idea of our paper was to evaluate the performance and robustness of EI across different datasets and settings. Nonetheless, we leave this as a direction for future work.

---

### Decision · Action_Editor_zQ5j · 2024-06-24

**Recommendation:** Accept as is

**Comment:**

Overall, the paper performs a thorough reproducibility study of the original Equal Improvability paper, confirming many of the original claims and extending the analysis to new datasets and settings. Reviewers appreciated the effort to communicate with the original authors, the well-written nature of the paper, and the (added) details needed for reproducing the reproducibility study. The paper meets expectations of the TMLR reproducibility paper.

**Audience:**

The paper would interest those focused on ML fairness. The paper provides valuable insights into the new fairness metric Equal Improvability, which could be beneficial for researchers and practitioners looking to implement or evaluate fairness metrics in dynamic decision-making systems.

**Claims And Evidence:**

The authors have reproduced the results of the original paper and extended the experiments to new datasets and settings (multiple sensitive attributes). Initially, there were some concerns regarding the lack of detailed experimental settings and justifications for certain decisions, but the authors have addressed these issues in their revision.